# Mapping Urban Forms Worldwide: An Analysis of 8910 Street Networks and 25 Indicators

**Qi Zhou** [1,*] **, Junya Bao** [1] **and Helin Liu** [2]

1   School of Geography and Information Engineering, China University of Geosciences, Wuhan 430074, China; estelle@cug.edu.cn
2   School of Architecture and Urban Planning, Huazhong University of Science and Technology, Wuhan 430074, China; hl362@hust.edu.cn
*   Correspondence: zhouqi@cug.edu.cn; Tel.: +86-15172382436

**Abstract:** Understanding urban form is beneficial for planners and designers to improve the built environment. The street network, as an essential element of urban form, has received much attention from existing studies. Recently, an open dataset containing 8910 global urban street networks and 25 different form indicators has been produced, but the urban forms of cities across the globe have rarely been recognized based on analyzing such a large dataset, which was the main purpose of our study. We employed correlation analysis, principal component analysis and hierarchical clustering methods for analyzing this dataset. We also compared the spatial pattern of clustering results with those using terrain and land-cover data. Results show that: (1) Most of these indicators are highly correlated with at least another indicator, and six principal components (i.e., size, terrain-variation, regularity, long-street, circuity and altitude) were found. (2) Seven clusters (i.e., regular, long-street, large size, irregular, varied-terrain, high-circuity and high-altitude) of cities were identified; cities of the same cluster can be spatially aggregated and also distributed across different regions. (3) Most of these clusters can be interpreted using terrain and land-cover data, which indicates that the urban forms of most cities across the globe are related to geographical factors. The clustering results may be used not only to compare street networks and their urban forms at a global scale but also to understand the formation and development of an urban street network.

**Keywords:** cities; form indicator; spatial pattern; geographical factor; global open data

## 1. Introduction

Urban form refers to the physical elements (e.g., streets, street blocks, plots and buildings) that structure and shape a city [1]. The physical form may have implications on various issues of a city, including public health and crime pattern [2–4], traffic safety and transportation system efficiency [5,6], energy use and urban heat island effect [7,8], economic development and pedestrian volume [9–11]. Thus, the research of urban form has received much attention from geographers, planners, engineers, architects and policy makers. As the backbone of a city, the street network has been viewed as an essential element of urban form. The 'form' or 'morphology' of a street network can refer to the geometry and topology of such a network [12], and it often varies in cities, probably due to reasons related to geographical, cultural and historical factors [13–16]. It is therefore desirable to identify the similarities and differences between different street networks in order to understand the urban forms of corresponding cities.

Extensive studies have proposed indicators and approaches for identifying the forms of different cities and their street networks. Schwarz [17] used various landscape metrics (e.g., number of patches and patch size) and population-related indicators for the analysis of urban form. Fang et al. [18] proposed using deep neural networks and planning guidance for the prediction of street configurations. Asami et al. [19] employed space-syntax-related indicators to identify traditional street networks in Turkish cities. Louf and

Barthelemy [20] used the area and shape of street blocks as indicators, and they divided 131 cities in the world into four clusters according to the size of area and the regularity of shape. Song et al. [21] compared 657 Chinese street networks based on two spatial scales (city-level and block-level). Tian et al. [22] designed network landscape indicators (e.g., evenness, richness and also size and shape) to cluster 100 cities worldwide. Boeing [23] used street orientations to examine 100 cities in the world and found that the United States and Canada cities are far more gird-like than those elsewhere. However, these studies have rarely discussed the relationship between urban form and geographical, cultural or historical factors.

Another branch is to investigate the reason(s) behind the physical form of a street network. This analysis is beneficial for planners and designers to understand not only how a street network has been shaped in the past but also how it will evolve in the future. Several studies have found that geographical factors (e.g., rivers, mountains and valleys) have great effects on the forms of street networks in Iranian cities [14,15]. Mohajeri et al. [24] also found the effects of adjacent shorelines on the spatial orientations of streets in Brazilian cities. Zhou et al. [25] employed spatial autoregressive models to quantitatively understand the relation between street network form and multiple geographical factors. They found from an analysis of Chinese cities that not only the terrain but also various land-cover types (e.g., cultivated lands and forests) can be significantly correlated with street orientations. Taubenböck et al. [26] found that similar urban forms are characterized by similar cultural, socio-economic, demographic or political conditions; nevertheless, they paid little attention to geographical factors. Despite a lot of existing studies, most have only used limited cities (e.g., 100–231) in the world as study areas, and few studies have involved a large number (e.g., almost 10,000) of cities across the globe in the analysis. With a large number of study areas, it may be possible to understand the similarities and differences between street networks and their cities in the spatial dimension. Moreover, although existing studies have found that geographical factors have great effects on the form of a street network, it is still not clear whether this finding is applicable to cities across the globe or in which geographical region(s) this finding is applicable.

Thanks to the open-source OSMnx software [27] and OpenStreetMap data (OSM, a free map database edited by global volunteers [28]), it has become easier to analyze and compare a large number of street networks in terms of multiple different indicators at a global scale. For instance, Moosavi [10] used a deep learning method to explore a large number of street networks in more than one million cities, towns and villages all over the world. He clustered these street networks into three groups, i.e., (1) villages and undeveloped areas; (2) the majority of towns and small cities; and (3) unique and very dense cities. Barrington-Leigh and Millard-Ball [29,30] proposed the Street-Network Disconnected index to detect eight typical street network types and used data on 46 million km of mapped streets worldwide to analyze a global time series of street network sprawl. However, the global pattern mapped by Moosavi [10] is related to urban development, and that mapped by Barrington-Leigh and Millard-Ball [29,30] is related to urban sprawl. Their clustering results can hardly be used to understand the relation between street network form and geographical factors on a global scale.

To fill the above research gaps, this study aimed to investigate the urban forms of cities across the globe based on analyzing a large number of street networks. This was achieved by employing recently produced open data that contained 8910 global urban street networks and 25 form indicators [12]. Moreover, as an extension of the existing study [25], we also investigated at a global scale whether such urban forms can be related to geographical factors, which was achieved by involving global terrain and land-cover data in the analysis.

This paper is structured as follows. Section 2 introduces global urban street network data and relevant form indicators, as well as terrain and land-cover data. Section 3 presents the framework to analyze the spatial pattern of street networks at a global scale. Sec-

tion 4 reports the analytical results. Sections 5 and 6 comprise discussion and conclusion, respectively.

## 2. Data and Methods

### 2.1. Global Urban Street Network Data

The global urban street network data, produced by Boeing [12] and published in September 2020, were used for analysis. There are several advantages of using the data. First, the data include street networks of 8910 cities across the globe (Table 1, each city is represented by an urban area derived from the publicly available Global Human Settlement Urban Centre Database, available online: https://ghsl.jrc.ec.europa.eu/ucdb2018Overview.php (accessed on 1 June 2022)). Second, in the datasets, there are not only geometric data but also dozens of form indicators for each street network. Specifically, in terms of the geometric data, the nodes and lines of each street network were acquired from OSM; in terms of the form indicators, there are 33 different indicators in total (Table 2), including the ID, country and city name of each street network, and also 25 different indicators that describe both geometric and topological forms of each street network (marked with '#' in Table 2). Most of these indicators were calculated based on the OSMnx software [26]. Third, the global urban street network data can be freely downloaded from the Harvard Dataverse (available online: https://dataverse.harvard.edu/dataverse/global-urban-street-networks/ (accessed on 1 June 2022)).

**Table 1.** Statistics of the 8910 cities across the globe.

| Region | Africa | Asia | Europe | North America | Oceania | South America |
|---|---|---|---|---|---|---|
| **Number of cities** | 1504 | 4935 | 1051 | 372 | 41 | 1007 |
| **Proportion (%)** | 16.88 | 55.39 | 11.80 | 4.18 | 0.46 | 11.30 |

More precisely, the data include five datasets for downloading and have approximately 80 GB in terms of file size. They are: (1) Global Urban Street Network Metadata: describes the information of the other four datasets; (2) Global Urban Street Network Indicators: contains all 33 indicators and corresponding values of each street network; (3) Global Urban Street Network GraphML: contains the geometric data of each street network in GraphML format; (4) Global Urban Street Network GeoPackages: contains the geometric data of each street network in GeoPackages format; (5) Global urban street network node/edge lists: contains the ID of each node and line of each street network in CSV format.

Despite the availability of different datasets, the geometric data (e.g., Global Urban Street Network GraphML) and various form indicators (i.e., Global Urban Street Network Indicators) are recorded separately. In order to integrate different datasets for the purpose of visualization and analyses, this study used a point or location to represent each street network and then connected this point to all indicators through the common field uc_id. The specific steps are listed in Appendix A. Finally, a point dataset including not only the centroid location of each urban street network but also corresponding values of all indicators in Table 2 can be produced. The produced new dataset integrates both geometric data and various indicators, it has a much smaller file size (around 11.6MB), and it has been made freely available to the public.

### 2.2. Terrain and Land-Cover Data

Furthermore, global terrain and land-cover data were also involved in the analysis. These data were used to investigate the relation between urban forms and different geographical factors. To be specific, the global terrain dataset (1km resolution) produced by Yale University was acquired (available online: http://www.earthenv.org/topography (accessed on 1 June 2022)) because it provided a number of topographic variables (e.g., mean of the elevation and standard deviation of the slope) for visualization and analysis [31]. The global land-cover dataset (300m resolution) produced by the European Space Agency was also acquired (available online: http://maps.elie.ucl.ac.be/CCI/viewer/download.php

(accessed on 1 June 2022)) because it not only includes multiple different land-cover types (e.g., agriculture and forest) but also has the recent year (2019). More importantly, both the terrain and land-cover datasets were freely available.

**Table 2.** Thirty-three indicators in the global urban street network data [12].

| ID | Indicator Name | Type | Description |
|---|---|---|---|
| 1 | country | string | Main country name |
| 2 | country_iso | string | Main country ISO 3166-1 alpha-3 code |
| 3 | core_city | string | Urban center core city name |
| 4 | uc_id | int | Urban center unique ID |
| 5 | circuity [#] | float | Ratio of street lengths to straight-line distances |
| 6 | elev_iqr [#] | float | Interquartile range of node elevations, meters |
| 7 | elev_mean [#] | float | Mean node elevation, meters |
| 8 | elev_median [#] | float | Median node elevation, meters |
| 9 | elev_range [#] | float | Range of node elevations, meters |
| 10 | elev_std [#] | float | Standard deviation of node elevations, meter |
| 11 | elev_res_mean | float | Average spatial resolution of elevation calculation |
| 12 | grade_mean [#] | float | Mean absolute street grade (incline) |
| 13 | grade_median [#] | float | Median absolute street grade (incline) |
| 14 | intersect_count [#] | int | Count of (undirected) edge intersections |
| 15 | intersect_count_clean [#] | int | Count of street intersections (after merging nodes within 10m of each other geometrically) |
| 16 | intersect_count_clean_topo [#] | int | Count of street intersections (after merging nodes within 10m of each other topologically) |
| 17 | k_avg [#] | float | Average node degree (undirected) |
| 18 | length_mean [#] | float | Mean street segment length, meters |
| 19 | length_median [#] | float | Median street segment length, meters |
| 20 | m [#] | int | Count of streets (undirected edges) |
| 21 | n [#] | int | Count of nodes |
| 22 | orientation_entropy [#] | float | Entropy of street network bearings |
| 23 | orientation_order [#] | float | Orientation order of street network bearings |
| 24 | prop_4way [#] | float | Proportion of nodes that represent 4-way street intersections |
| 25 | prop_3way [#] | float | Proportion of nodes that represent 3-way street intersections |
| 26 | prop_deadend [#] | float | Proportion of nodes that represent dead-ends |
| 27 | straightness [#] | float | 1/circuity |
| 28 | uc_names | string | List of city names within this urban center (GISCO) |
| 29 | world_region | string | Major geographical region (UN WUP) |
| 30 | world_subregion | string | Geographical region (UN WUP) |
| 31 | resident_pop [#] | int | Total resident population in 2015 (GHS) |
| 32 | area [#] | float | Area within urban center boundary polygon, km$^2$ (GHS) |
| 33 | built_up_area [#] | float | Built-up surface area in 2015, km$^2$ (GHS) |

'#' denotes the indicator used for analyzing global urban street networks.

## 3. Methods

### 3.1. Correlation Analysis

Several existing methods may be used to analyze the produced point dataset [17,20,22]. For instance, Schwarz [17] proposed selecting a few indicators from a larger set for analyzing the urban form, which has been applied to 231 European cities. A similar idea has also been employed by Tian et al. [22] and applied to 100 cities worldwide. Louf and Barthelemy [20] applied a hierarchical clustering method to 131 cities in the world. By referring to these studies, we used an analytical framework that involved four steps, i.e., (1) correlation analysis, (2) principal component analysis, (3) clustering analysis and (4) clustering results interpretation.

The point dataset has 33 indicators in total, including 25 indicators that describe various forms of each street network. Some indicators may be strongly correlated. For instance, the indicator straightness is an inverse of the circuity (Table 2). This indicates that the spatial patterns of using some indicators are similar or even the same. The purpose of the correlation analysis was to investigate: (1) whether any of the 25 indicators were highly correlated and (2) which of them were highly correlated.

Specifically, the Pearson correlation coefficient was employed to calculate the correlations among indicators [32], and it assigned a value between −1 and 1, where 0 denotes no

correlation, 1 denotes total positive correlation, and −1 denotes total negative correlation. A correlation value above 0.6 (or below −0.6) indicates that at least a (moderately) strong positive (or negative) relationship exists between two indicators [32].

### 3.2. Principal Component Analysis

Because some of the 25 indicators may be highly correlated, it was necessary to reduce such a large set of indicators into a smaller set [17], which was then used in further analysis. The principal component analysis (PCA) was employed [33]; the method could transform the original 25 indicators into several new principal components (or PCs) with minimal loss of information. Each principal component (PC) has an eigenvalue to indicate the total amount of variance that can be explained by this PC. Commonly, PCs whose eigenvalues are greater than one are kept, which means that these PCs can account for more variance than any original indicator did.

A 'varimax rotation' was applied to the kept PCs because it makes all the 25 indicators strongly correlate with only a single PC, thus making the results more interpretable [33]. In order to understand the meaning of each PC, the correlations between all the kept PCs and the original 25 indicators were calculated. A larger correlation coefficient indicates a stronger correlation between a pair of indicator and PC.

### 3.3. Clustering Analysis

Based on the PCs found by principal component analysis, this study further employed the hierarchical clustering method [34] to group the 8910 urban street networks into different clusters. The hierarchical clustering method was used for several reasons: First, this method can provide a hierarchical tree or dendrogram, which records the relationship of merges or splits during the clustering processing. With the dendrogram, an appropriate number of clusters may be visually determined. Second, this method has been applied to analyze street network form in several existing studies [20,21,35].

The tenet of the hierarchical clustering method is firstly to treat each street network as an individual cluster. Then, the 'distance' between each pair of street networks can be calculated based on multiple factors. Ward's method [36], aiming to minimize the error sum of squares (ESS), was used to calculate the 'distance'. After that, the pair of street networks with the smallest 'distance' was grouped into the same cluster. The process of grouping was repeated until all street networks were grouped into the same cluster.

Furthermore, the optimal number of clusters can be determined using the elbow method. The principle of this method is to calculate the within-cluster sum of squared errors (WCSS) for different numbers of clusters and to identify the number of clusters that the WCSS dramatically decreases as the optimal one [37]. After that, the clustering results of 8910 cities could be visualized on a map, after which step the similarities and differences between various cities could be observed.

### 3.4. Clustering Results Interpretation

It is needed to explain the clustering results, that is, to explain the characteristics of street networks in each cluster. First of all, this can be achieved in two ways called visual inspection and quantitative analysis. 1) Visual inspection: The street networks of some typical cities are picked and visualized; after that, the characteristics of these street networks are visually observed. In this study, two or three street networks were picked for each cluster. 2) Quantitative analysis: The cities of each cluster are selected; then, the median of the selected cities is calculated in terms of each PC. Finally, the medians of different PCs are compared for each cluster. It means that the identified PCs are used to quantitatively understand the different clustering results.

Moreover, the spatial patterns of the clustering results were visually compared with those obtained from both global terrain and land-cover data in order to investigate the relation between various clusters and different geographical factors.

## 4. Results and Analyses

### 4.1. Correlation Analysis

The correlations among all pairs of 25 indicators are shown in Figure 1. The positive and negative correlations are highlighted in red and blue, respectively. The higher the correlation, the darker the color.

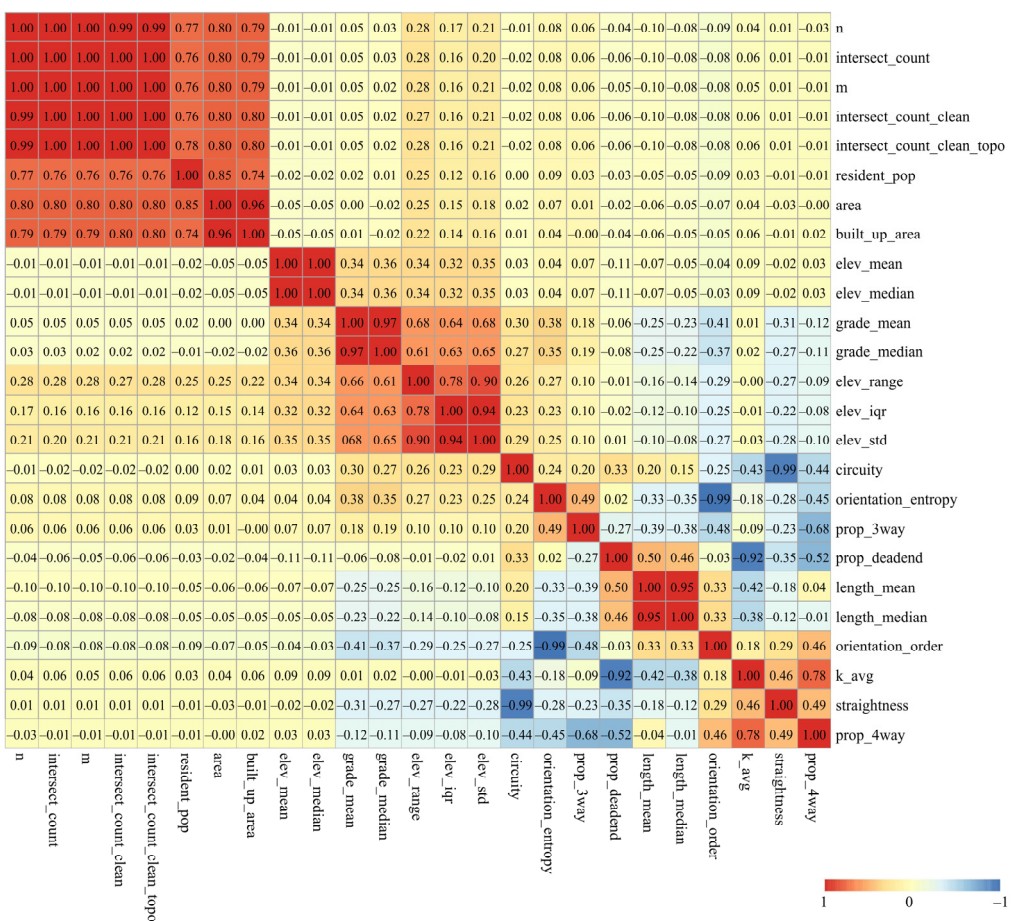

**Figure 1.** Correlations among all pairs of 25 indicators. The *p*-value is smaller than 0.05, while the correlation is larger than 0.02 or smaller than −0.02.

Figure 1 shows that: The correlation varies from 1.00 (a perfectly positive correlation) to −0.99 (a perfectly negative correlation). Most indicators have a strong correlation (larger than 0.6 or smaller than −0.6) with at least another indicator. Some pairs of indicators have a strong positive correlation, e.g., elev_mean and elev_median (1.00), grade_mean and grade_median (0.97), length_mean and length_median (0.95), k_avg and prop_4way (0.78). Some pairs of indicators have a strong negative correlation, e.g., circuity and straightness (−0.99), orientation_entropy and orientation_order (−0.99), prop_deadend and k_avg (−0.92). In particular, some indicators have a strong positive correlation with each other, i.e., seven indicators: n, intersect_count, m, intersect_count_clean, intersect_count_clean_topo, resident_pop, area and built_up_area. This is because a larger street network usually has more streets (m), nodes (n) and intersections (intersect_count) and also a larger size of built-up areas (built_up_area) and more population (resident_pop). There is also a strong positive correlation among five indicators, i.e., grade_mean, grade_median, elev_range, elev_iqr and elev_std. These indicators are all related to terrain. It should also be noted that the terrain-related indicators (grade_mean and grade_median) have a moderate correlation (e.g., 0.35−0.38) with orientation_entropy and orientation_order, which indicates the relationship between urban form and geographical factors. More importantly,

the above findings verify that most of the 25 indicators are highly correlated, and thus it is necessary to reduce these indicators into a smaller set.

### 4.2. Principal Component Analysis

Six PCs were found, and their eigenvalues are all greater than one. These PCs can explain about 86% of the variance in the whole dataset. Specifically, Table 3 shows the variance that can be explained with each PC and also the correlations between each PC and the original 25 indicators. We highlight the correlation coefficient with an underscore if the value is higher than 0.6 or smaller than −0.6.

**Table 3.** Correlations between each of 6 PCs and the original 25 indicators.

| Original Indicator | PC | | | | | |
|---|---|---|---|---|---|---|
| | 1 | 2 | 3 | 4 | 5 | 6 |
| | % of variance | | | | | |
| | 30.06 | 20.55 | 14.64 | 10.93 | 5.88 | 4.40 |
| area | 0.899 | 0.053 | 0.007 | −0.008 | 0.034 | −0.059 |
| built_up_area | 0.881 | 0.042 | 0.030 | −0.030 | 0.031 | −0.050 |
| intersect_count | 0.974 | 0.066 | −0.040 | −0.029 | −0.020 | 0.013 |
| intersect_count_clean | 0.976 | 0.066 | −0.037 | −0.030 | −0.019 | 0.013 |
| intersect_count_clean_topo | 0.976 | 0.067 | −0.037 | −0.030 | −0.019 | 0.013 |
| m | 0.976 | 0.067 | −0.037 | −0.027 | −0.021 | 0.012 |
| n | 0.975 | 0.071 | −0.044 | −0.015 | −0.020 | 0.011 |
| resident_pop | 0.844 | 0.048 | −0.025 | 0.002 | 0.000 | −0.019 |
| elev_iqr | 0.115 | 0.881 | −0.037 | 0.018 | 0.041 | 0.106 |
| elev_range | 0.234 | 0.850 | −0.072 | −0.003 | 0.081 | 0.131 |
| elev_std | 0.156 | 0.914 | −0.031 | 0.039 | 0.092 | 0.126 |
| grade_mean | −0.034 | 0.841 | −0.219 | −0.108 | 0.139 | 0.116 |
| grade_median | −0.060 | 0.817 | −0.210 | −0.121 | 0.120 | 0.151 |
| orientation_entropy | 0.037 | 0.282 | −0.840 | 0.024 | 0.017 | −0.079 |
| orientation_order | −0.042 | −0.306 | 0.833 | −0.033 | −0.024 | 0.085 |
| prop_4way | −0.011 | 0.038 | 0.705 | −0.519 | −0.318 | −0.061 |
| prop_3way | 0.037 | −0.057 | −0.751 | −0.208 | 0.279 | 0.156 |
| k_avg | 0.038 | 0.028 | 0.303 | −0.888 | −0.212 | 0.029 |
| length_mean | −0.056 | −0.121 | 0.504 | 0.713 | 0.159 | 0.004 |
| length_median | −0.042 | −0.094 | 0.518 | 0.689 | 0.105 | 0.017 |
| prop_deadend | −0.030 | 0.009 | −0.049 | 0.919 | 0.083 | −0.093 |
| circuity | −0.014 | 0.230 | −0.127 | 0.234 | 0.918 | −0.024 |
| straightness | 0.011 | −0.225 | 0.181 | −0.253 | −0.905 | 0.031 |
| elev_mean | −0.031 | 0.278 | 0.005 | −0.046 | −0.022 | 0.950 |
| elev_median | −0.033 | 0.274 | 0.006 | −0.046 | −0.022 | 0.951 |

The underscore denotes that the value is higher than 0.6 or smaller than −0.6.

Table 3 shows that:

PC1 can explain about 30% of the variance in the dataset. This PC has a strong positive correlation with eight indicators, including area, built_up_area, intersect_count, intersect_count_clean, intersect_count_clean_topo, m, n and resident_pop. All these indicators are related to the size of a city and/or its street network. A high value of this PC indicates a relatively large size of a corresponding city. Thus, this PC is denoted as 'Size'.

PC2 can explain about 20% of the variance in the dataset. This PC has a strong positive correlation with five indicators, including elev_iqr, elev_range, elev_std, grade_mean and grade_median. All these indicators are used to describe the terrain of a city. A high value of this PC indicates that a city is located in a varied terrain. Thus, this PC is denoted as 'Terrain-variation'.

PC3 can explain about 15% of the variance in the dataset. This PC has a strong positive correlation with the two indicators orientation_order and prop_4way but a strong negative correlation with the other two indicators orientation_entropy and prop_3way.

These indicators can be used to describe the regular/irregular pattern of an urban street network. A high value of this PC indicates a grid/regular street network pattern, and a low value indicates a relatively irregular street network pattern. Thus, this PC is denoted as 'Regularity'.

PC4 can explain about 11% of the variance in the dataset. This PC has a strong positive correlation with the three indicators length_mean, length_median and prop_deadend but a strong negative correlation with the indicator k_avg. These indicators are related to not only the connectivity of a street network but also the length of each street segment. A high value of this PC indicates a street network with longer street segments and more dead-end roads, and a low value indicates a street network with relatively short street segments and few dead-end roads. Thus, this PC is denoted as 'Long-street' for brevity.

PC5 can explain about 6% of the variance in the dataset. This PC has a strong positive correlation with the indicator circuity but a strong negative correlation with the indicator straightness. These indicators can be used to describe whether most street segments are curved or straight. A high value of this PC indicates that most street segments are curved. Thus, this PC is denoted as 'Circuity'.

PC6 can explain about 4% of the variance in the dataset. This PC has a strong positive correlation with the indicators elev_mean and elev_median. These indicators are used to describe the altitude at which a city is located. A high value of this PC indicates that a city is located at a high altitude. Thus, this PC is denoted as 'Altitude'.

### 4.3. Clustering Analysis

The 8910 urban street networks were further clustered based on the six PCs in Table 3. The hierarchical clustering method was implemented in the data mining software IBM SPSS Statistics (version 22), after which step, seven clusters were identified and their spatial patterns were visualized in the geographic information system software ArcGIS (version 10.6) [38]. Both the hierarchical diagram and the spatial patterns of these clusters are shown in Figure 2.

The 8910 urban street networks can be grouped into seven clusters (Figure 2a–c), colored red (Cluster I), gray (Cluster II), black (Cluster III), green (Cluster IV), purple (Cluster V), yellow (Cluster VI) and blue (Cluster VII). Figure 2c shows that 70% ((1866 + 4415)/8910 = 70.5%) of cities are clustered as either Cluster I or Cluster IV, and almost 50% (4415/8910 = 49.6%) of cities are clustered as Cluster IV.

Moreover, cities of the same cluster can not only be distributed in different countries and regions (Appendix B) but also be spatially aggregated. As an example, most cities in Europe and India are clustered into Cluster IV; most cities in Central United States, Argentina and Eastern China are clustered into Cluster I. The possible reasons are discussed in Section 4.4.

### 4.4. Interpretation of Clustering Results

For the purpose of interpreting each cluster, not only 16 cities were picked up and visualized for visual inspection (Figure 3), but also the medians of six PCs were calculated for cities in each cluster and plotted for quantitative analysis (Figure 4). To be specific, the cities in Figure 3 were chosen because they are not only located in some typical countries and regions mentioned above but also close to corresponding cluster centroids. In Figure 4, a median larger (or smaller) than zero indicates a positive (or negative) correlation between the corresponding pair of PC and cluster. The larger the absolute of such a median, the stronger the correlation is.

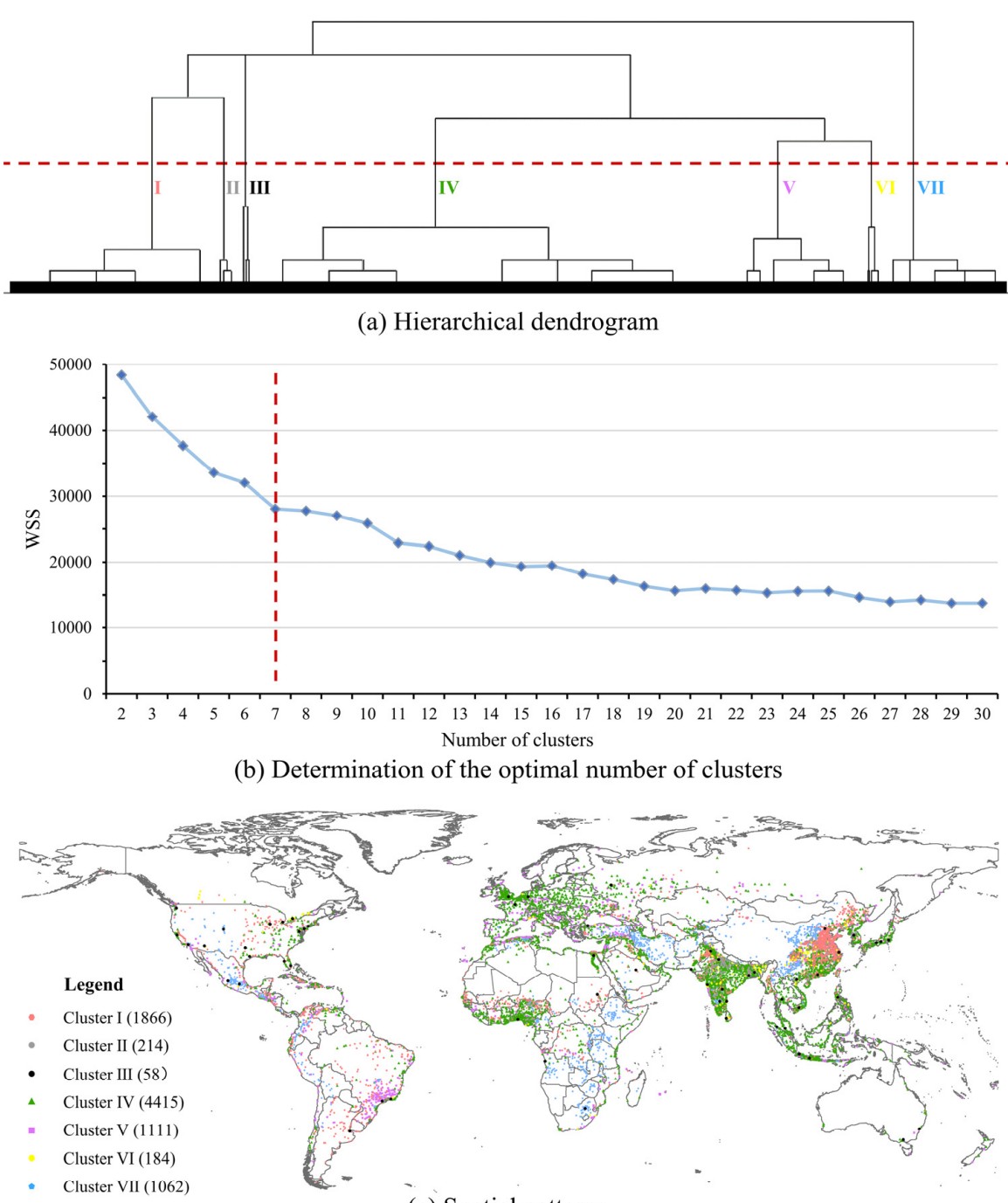

(a) Hierarchical dendrogram

(b) Determination of the optimal number of clusters

(c) Spatial patterns

**Figure 2.** A hierarchical clustering of global urban street networks based on six PCs.

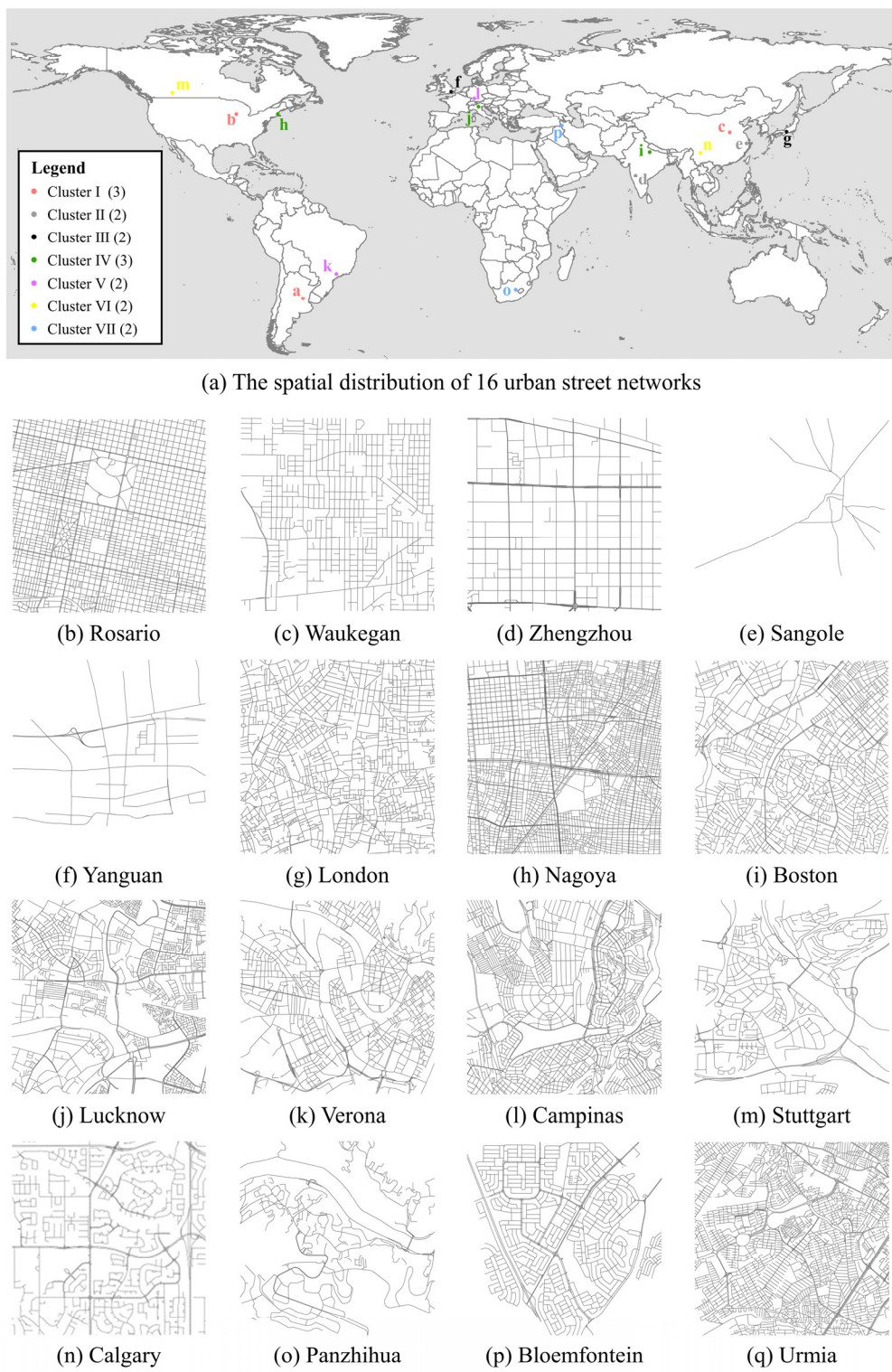

(a) The spatial distribution of 16 urban street networks

| | | | |
|---|---|---|---|
| (b) Rosario | (c) Waukegan | (d) Zhengzhou | (e) Sangole |
| (f) Yanguan | (g) London | (h) Nagoya | (i) Boston |
| (j) Lucknow | (k) Verona | (l) Campinas | (m) Stuttgart |
| (n) Calgary | (o) Panzhihua | (p) Bloemfontein | (q) Urmia |

**Figure 3.** The detailed view of 16 typical urban street networks (each was visualized in a 5 km × 5 km grid).

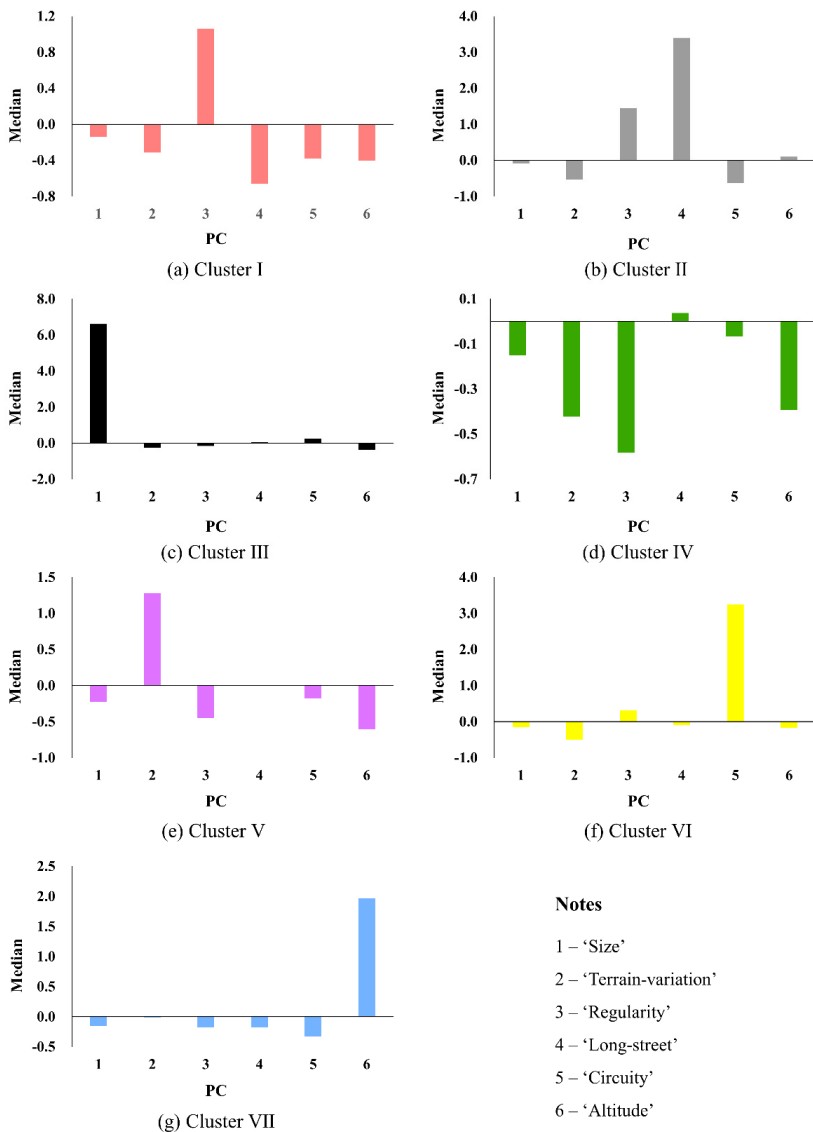

**Figure 4.** The medians of six PCs for cities in each cluster.

We can see from Figures 2–4 that:

(1) Cluster I ('Regular'): The cities in this cluster are mostly located in Central United States, Argentina, Brazil and Eastern China. Most of the cities have a relatively high value in terms of PC3 ('Regularity'). Thus, the cities in this cluster are characterized by a regular street network pattern, e.g., Rosario (Figure 3b), Waukegan (Figure 3c) and Zhengzhou (Figure 3d).

(2) Cluster II ('Long-street'): The cities in this cluster are mostly located in India and China. Most of the cities have a relatively high value in terms of PC4 ('Long-street'). Thus, the cities in this cluster are characterized by long street segments, e.g., Sangole (Figure 3e) and Yanguan (Figure 3f).

(3) Cluster III ('Large-size'): There are only 58 cities in this cluster. Most of the cities have a relatively high value in terms of PC1 ('Size'). Thus, the cities in Cluster I, e.g., London (Figure 3g) and Nagoya (Figure 3h), are characterized by a large size.

(4) Cluster IV ('Irregular'): The cities in this cluster are mostly located in Eastern United States, Europe, Western and Northern Africa, Southeast Asia, United Kingdom, Egypt, India and Japan. Most of the cities have a relatively low value in terms of PC2 ('Terrain-variation'), PC3 ('Regularity') and PC6 ('Altitude'). Thus, the cities in this cluster are located at a flat terrain and low altitude, and they are also characterized by

an irregular street network pattern, e.g., Boston (Figure 3i), Lucknow (Figure 3j) and Verona (Figure 3k).

(5) Cluster V ('Varied-terrain'): The cities in this cluster are mostly located in North America, South America and Europe. Most of the cities have a relatively high value in terms of PC2 ('Terrain-variation') and a relatively low value in terms of PC3 ('Regularity') and PC6 ('Altitude'). Thus, the cities in Cluster V are located at a varied terrain and a low altitude, and they are also characterized by an irregular street network pattern, e.g., Campinas (Figure 3l) and Stuttgart (Figure 3m).

(6) Cluster VI ('High-circuity'): The cities in this cluster are mostly located in Canada, India and China. Most of the cities have a relatively high value in terms of PC5 ('Circuity'). Thus, the cities are characterized by curved street segments, e.g., Calgary (Figure 3n) and Panzhihua (Figure 3o).

(7) Cluster VII ('High-altitude'): The cities in this cluster are mostly located in Western United States, Mexico, Western South America, Eastern and Southern Africa, Turkey, Iran and Western China. Most of the cities have a relatively high value in terms of PC6 ('Altitude'). Thus, the cities in this cluster are located at a high altitude, e.g., Bloemfontein (Figure 3p) and Urmia (Figure 3q).

Furthermore, Figure 5 compares the spatial patterns of various clustering results with those of different geographical factors. Specifically, the four typical clusters (I, IV, V and VII) are visualized because almost 95% (8454/8910) of all the street networks were clustered as one of these types. Two land-cover types (agriculture and forest) and two terrain variables (mean of the elevation and standard deviation of the slope) are visualized because an existing study has reported that these types and variables can be significantly correlated with street network form [25]. We can see from Figure 5 that:

(1) Cluster I ('Regular'): The cities in this cluster are mostly located in regions with a relatively high proportion of agriculture lands, e.g., Central United States ('A'), Argentina ('B') and North China ('C') (Figure 5b). In a general sense, agricultural lands are more likely to be located in alluvial plains (Appendix C), particularly in ancient times when engineering technology was not well developed and cities were emerging with the boom of agriculture. Alluvial plains allow the possibility of developing regular road networks, which can avoid too many costs in labor and investment and engineering challenges. Comparatively, this kind of regular network system is simple and easy to build. It was further massively reproduced after the industrial revolution which had tremendously promoted the rapid development of cities and taken mass and standard production as the key principle. As David and Tong put it, during the Industrial Revolution, cities could be interpreted as being made up of standardized components, which had changed the typology and terrain rooted in geography or culture [39]. These developments then left us the legacy of regular street networks as we see in the regions with a relatively high proportion of agriculture lands.

(2) Cluster IV ('Irregular'): The cities in this cluster are mostly located in regions with a relatively high proportion of forests, e.g., Europe ('D'), Southern China ('E') and Southeast Asia ('F') (Figure 5d and Appendix D). From the physical geography perspective, forests are more likely to be observed in regions with hills or mountains. It means that these regions characterized by a higher density of forests have a much more complex terrain than those in the alluvial plains. Therefore, cities in these regions have to conquer a series of engineering challenges. Yet, cities are the product of historical processes, not only because of their physical nature but also because of their cultural attributes [40]. In the early development stage of cities, city builders had to find solutions from nature and not modern engineering technologies as we see today. In this sense, it was reasonable to follow the natural terrains while developing the city road networks [41]; this produced irregular network systems like what we see today. In addition to the influence of physical geography, the irregularity can also be partly attributed to cultural elements [40,42]. For instance, the irregularity of street networks

in European cities was the legacy of the medieval time during which cities/towns experienced natural evolution with no intentional planning. This kind of idea was largely inherited by later developers, and the idea of natural irregular patterns was also brought to India by early colonies from Europe [43]. Nowadays, despite the massive influence of modern principles and mechanic aesthetics in planning, we still can see these traditions and legacies in these regions.

(3) Cluster V ('Varied-terrain') and Cluster VII ('High-altitude'): The cities in these two clusters are all related to terrain. As an example, those located in Brazil ('G'), Europe ('H') and Southeast Asia ('I') are characterized by a varied terrain, or a relatively large standard deviation of the slope (Figure 5f). As another example, those located in Mexico ('J'), Eastern Africa ('K'), Iran ('L') and Southwest China ('M') are characterized by a high altitude or a relatively large mean of the elevation (Figure 5h). This verifies the results found in Figure 4. In addition, the cities in Cluster V are also characterized by an irregular pattern, which also verifies that the terrain can have an impact on the regularity of a street network.

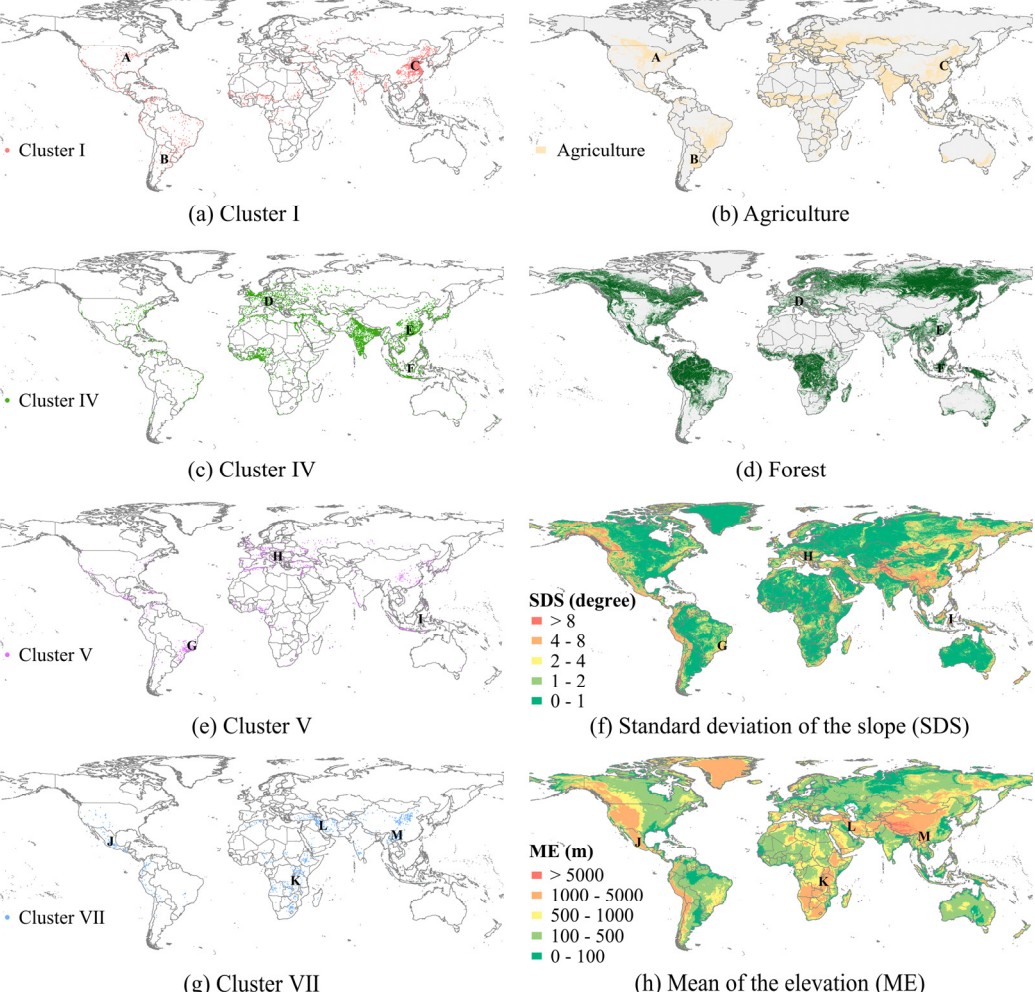

**Figure 5.** Comparison of the spatial patterns of the four typical clusters, i.e., (**a**) Cluster I, (**c**) Cluster IV, (**e**) Cluster V and (**g**) Cluster VII, with those of the four geographical factors, i.e., (**b**) agriculture, (**d**) forest, (**f**) standard deviation of the slope and (**h**) mean of the elevation.

Although the results in Figure 5 indicate that the urban forms of most cities can be related to geographical factors (including terrain variables and various land-cover types), there are some exceptions. Typically, cities in India are mostly located in regions with a relatively large proportion of agriculture lands rather than forests, but they are mostly

characterized by an irregular street network pattern. This is probably due to the effect(s) of European colonization in the history of India [43], which indicates that historical and/or cultural factors may have greater impacts on the urban form in this region.

## 5. Discussion

This study analyzed 8910 global urban street networks based on 25 form indicators. Seven typical clusters were identified, and they include street networks with a grid/regular pattern (Cluster I), with long street segments (Cluster II), with a large size (Cluster III), with an irregular pattern (Cluster IV), located at a varied terrain (Cluster V), with curved street segments (Cluster VI) and located at a high altitude (Cluster VII).

### 5.1. Comparing with Existing Studies

First of all, most existing studies have used areal indicators (e.g., patches, blocks or cells) for the clustering of urban street networks [17,20,22]. However, in this study, we employed indicators related to street nodes (e.g., intersect_count and prop_4way) and street segments (e.g., length_mean and circuity), which can be viewed as point and linear indicators, respectively. Thus, our clustering results are different from those found in existing studies. More importantly, most of the existing studies have only focused on proposing indicators and approaches for the identification of urban form, but in our study, the relation between various street network forms and different geographical factors was also investigated. Thus, our study can be viewed as an extension and supplement of existing studies.

Furthermore, a large number (8910) of global street networks were analyzed, much more than those (e.g., 100−231) in most existing studies [17,20,22]. More importantly, with such a large sample size, it is possible to investigate the similarities and differences in street network form not only across different countries but also within a country. As an example, we found that the street networks in the United States may be characterized by an irregular pattern (e.g., Boston), but they were mostly recognized as a grid/regular pattern in previous studies [23]. Thus, the clustering results also extend our understanding(s) of street network forms worldwide. Moreover, although several studies have also analyzed a large number of street networks [10,29,30], their clustering results are related to urban development [10] and urban sprawl [29,30]. In contrast, our clustering results are useful to understand the relation between street network forms and geographical factors on a global scale.

In addition, a new dataset was produced for the global-scale analysis. There are several advantages of using this dataset: (1) the dataset has a much smaller file size (11.6MB) compared with those (80GB in total) produced by Boeing [12]; (2) the dataset has been saved in Shapefile format, which is applicable to most GIS software programs (e.g., ArcGIS and QGIS); (3) in this dataset, the centroid location of each urban street network has been connected to its 33 form indicators, and there is no need to develop an additional tool for the data processing; (4) the dataset has also been made for public use. To sum up, the new dataset is free and easy to use, and it is an alternative for mapping and analyzing the spatial pattern of urban form at a global, regional and/or national scale.

### 5.2. Implications

First of all, some existing studies have found that geographical factors have great effects on the urban form [14,15,24,25]; some others have highlighted the effects of historical and cultural factors [12,16,26]. However, most of these studies have only been carried out based on analyzing one city or limited cities in a region. In this study, we found from analyzing a much larger number (8910) of urban street networks that the urban forms of most cities across the globe can be related to geographical factors. In other words, from the very starting point, city planning and design had followed nature, which as a result had left the legacy as what we revealed in this research. The key reason why nature had such an evident impact on early city development was the various geographical

terrains and the limitation of engineering technology that defined what city builders could do. Although the industrial revolution and the development of modern technology had presented human beings with sharp tools to overcome the geographical constraints, urban planning and design with nature still had their footprint on cities across the globe [44]. Moreover, our finding may be beneficial for planners and designers to understand how an urban street network has been shaped in the past and also how it will evolve in the future. This may be achieved by involving terrain and land-cover data to simulate the evolution of a street network.

Furthermore, the clustering results may also be used to investigate the relation between street network forms and socio-economic developments. As an example, we divided the 8910 cities into four income groups: high income, upper-middle income, lower-middle income and low income. To be specific, all the cities of a country were divided into the same group according to the classification of World Bank (available online: https://datahelpdesk.worldbank.org/knowledgebase/articles/906519-world-bank-country-and-lending-groups (accessed on 1 June 2022)). For each group, we calculated the proportion of cities in terms of each cluster (Figure 6). Figure 6 shows that in the high-income group, there is a relatively higher percentage of large-size cities (Cluster III) than that in the low-income group, but there is a relatively lower percentage of cities with long street segments (Cluster II) and at a high altitude (Cluster VII) than that in the lower-middle- and low-income groups. Thus, the difference in street network forms may have implications for socio-economic developments. More importantly, the finding may also be interesting to planners and local officers because the street networks of some lower-middle- and/or low-income countries may be improved (e.g., by increasing street network density) to promote socio-economic developments.

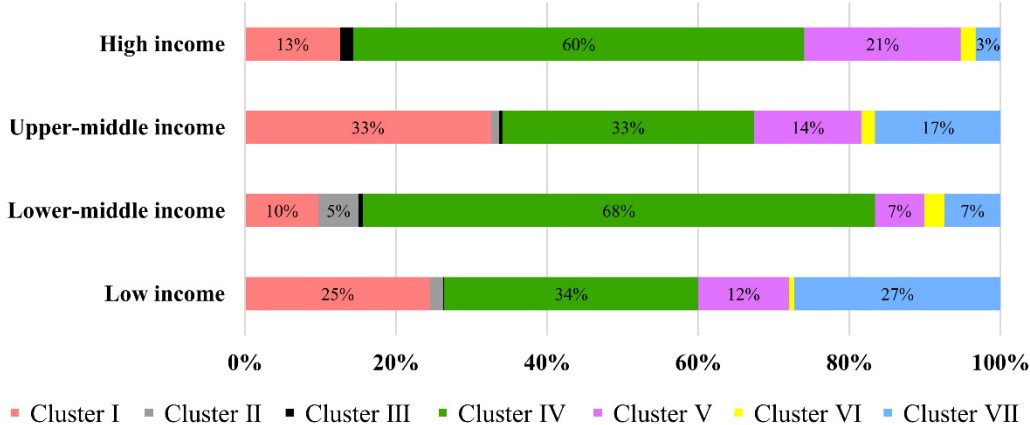

**Figure 6.** The proportions of cities for the seven clusters in each income group.

*5.3. Limitations of This Study*

There are also limitations of this study. First of all, the principal component analysis was used to transform the original 25 form indicators into a few PCs, based on which the hierarchical clustering was employed for clustering 8910 cities into seven typical clusters. However, both the PCs and clusters may be different if using different analysis and clustering methods. This study did not compare different methods because there is a lack of a benchmark to quantitatively evaluate which method can perform the best. In future work, however, it is still worthwhile to try different methods. Moreover, in addition to the 25 form indicators, it is also interesting to combine other indicators (e.g., fractal dimension, compact ratio and shape factor [21]) into analysis in order to investigate the spatial pattern of global urban street networks from different perspective(s).

Furthermore, most of the indicators (Table 2) were calculated based on OSM data [12]. However, many concerns have been raised about the quality of the data [28,45,46]. Nevertheless, Barrington-Leigh and Millard-Ball [47] found at a global level that the OSM road

data are more than 80% complete. A recent study [48] has also reported that major roads have already been mapped well. More importantly, there still is a lack of authoritative (e.g., produced by mapping agencies or commercial companies) and free road network data in most countries and regions. Therefore, the use of OSM data may currently be the only choice for a global study.

Last but not least, although we found that the urban forms of most cities across the globe are related to geographical factors (Figure 5), some exceptions were also found, e.g., in India. This indicates that other factors (e.g., historical and/or cultural factors) may also have impacts on the urban form, as reported in several existing studies [13,16,26]. However, there still is a lack of global open data for historical and/or cultural factors, which were therefore not considered in our study. Nevertheless, it is needed to combine different factors (e.g., geographical, historical and cultural factors) to investigate the urban form of each city and its street network; in addition, it is also worthwhile to investigate which is the dominant factor for each city because such a factor may vary with different cities. Furthermore, how to utilize the clustering results for urban planning and design is still ongoing work. With such a large number (8910) of street networks, it is possible to quantitatively understand the relation between urban forms and various urban issues (e.g., public health, social justice, climate change and energy use [1,49]), which may be beneficial for planners and designers to improve our built environment.

## 6. Conclusions

This study investigated the similarities and differences between 8910 global urban street networks based on 25 different form indicators. First of all, a new dataset was produced to perform such a global-scale analysis. Then, we used an analytical framework involving several steps. First, the correlations among multiple indicators were calculated, and the principal component analysis was employed to transform the 25 indicators into several PCs. Second, a hierarchical clustering method was employed to cluster these PCs. Third, the clustering results were interpreted based on quantitative analysis and also visually comparing with global terrain and land-cover data. Results show that:

First, there is a strong correlation (larger than 0.6 or smaller than −0.6) among most of the 25 form indicators, and these indicators can be reduced into six PCs. Second, the six PCs can be clustered into seven typical clusters. Specifically, the seven clusters indicate street networks and corresponding cities of different characteristics, i.e., with a grid/regular pattern (Cluster I), with long street segments (Cluster II), with a large size (Cluster III), with an irregular pattern (Cluster IV), located at a varied terrain (Cluster V), with curved street segments (Cluster VI) and located at a high altitude (Cluster VII). Cities of the same cluster can not only be spatially aggregated but also be distributed across different regions. Third, most of these clusters may be interpreted using terrain variables (mean of the elevation and standard deviation of the slope) and land-cover types (agriculture and forest), which indicates that the urban forms of most cities across the globe are related to geographical factors. However, this is not the case for some regions (e.g., India) in the world, probably due to the impacts of historical and/or cultural factors.

In future work, first, other analysis and clustering methods will be considered to check whether the clustering results may be different. Second, the analytical framework will be applied to specific countries and regions for analysis. Third, other measures will be combined to understand the similarities and differences between urban forms of cities across the globe and also to quantitatively understand the relation between urban forms and various urban issues in order to improve our built environment.

**Author Contributions:** Conceptualization, Methodology, Writing—Original Draft Preparation, Qi Zhou; Data Curation, Visualization, Analysis and Validation, Junya Bao; Analysis, Writing—Reviewing and Editing, Helin Liu. All authors have read and agreed to the published version of the manuscript.

**Funding:** The work was supported by the National Natural Science Foundation of China (grant no. 41771428) and Fundamental Research Funds for the Central Universities, China University of Geosciences (Wuhan) (grant no. CUGESIW1801).

**Institutional Review Board Statement:** Not applicable.

**Informed Consent Statement:** Not applicable.

**Data Availability Statement:** The dataset of this study is openly available on figshare at https://figshare.com/s/93a8c940606076f1d560 (accessed on 1 June 2022).

**Conflicts of Interest:** The authors declare no conflict of interest.

## Appendix A

The steps to produce a point dataset including both geometric data and 33 indicators.

Step 1: Convert the geometric data in GraphML format into those in Shapefile format in OSMnx software. The Shapefile format is a vector data format, which is compatible with most geographic information system software programs (e.g., ArcGIS and QGIS).

Step 2: Import the geometric data in Shapefile format into ArcGIS software and create the convex hull (the smallest convex polygon containing all the nodes and lines) of each urban street network.

Step 3: Calculate the centroid point of each convex hull, and this point is used to represent the location of each urban street network.

Step 4: Develop a Python-based tool to read the uc_id from the file name of each urban street network and to write the uc_id into the attribute table of the corresponding centroid point.

Step 5: Use the uc_id field to connect each centroid point to the corresponding values of all indicators.

## Appendix B

Typical countries and regions in the world, for illustrating various forms of global urban street networks.

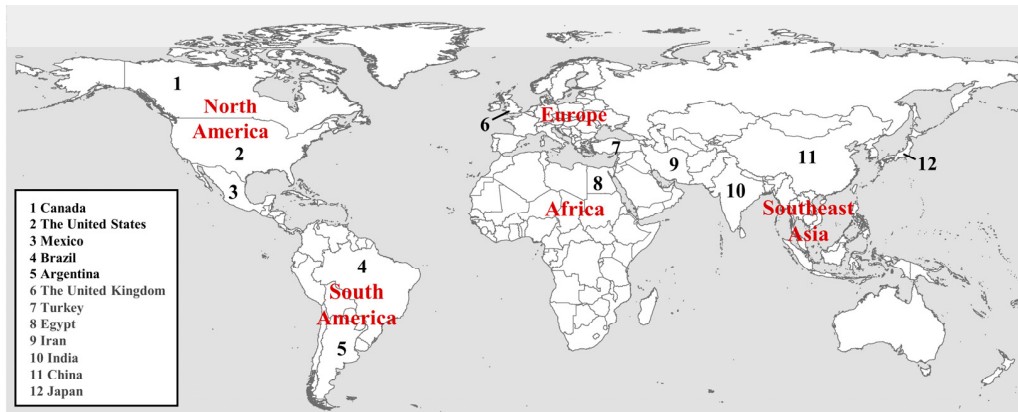

## Appendix C

Comparing the spatial pattern of different clusters and land-cover maps, for geographical regions with a relatively large percentage of agriculture.

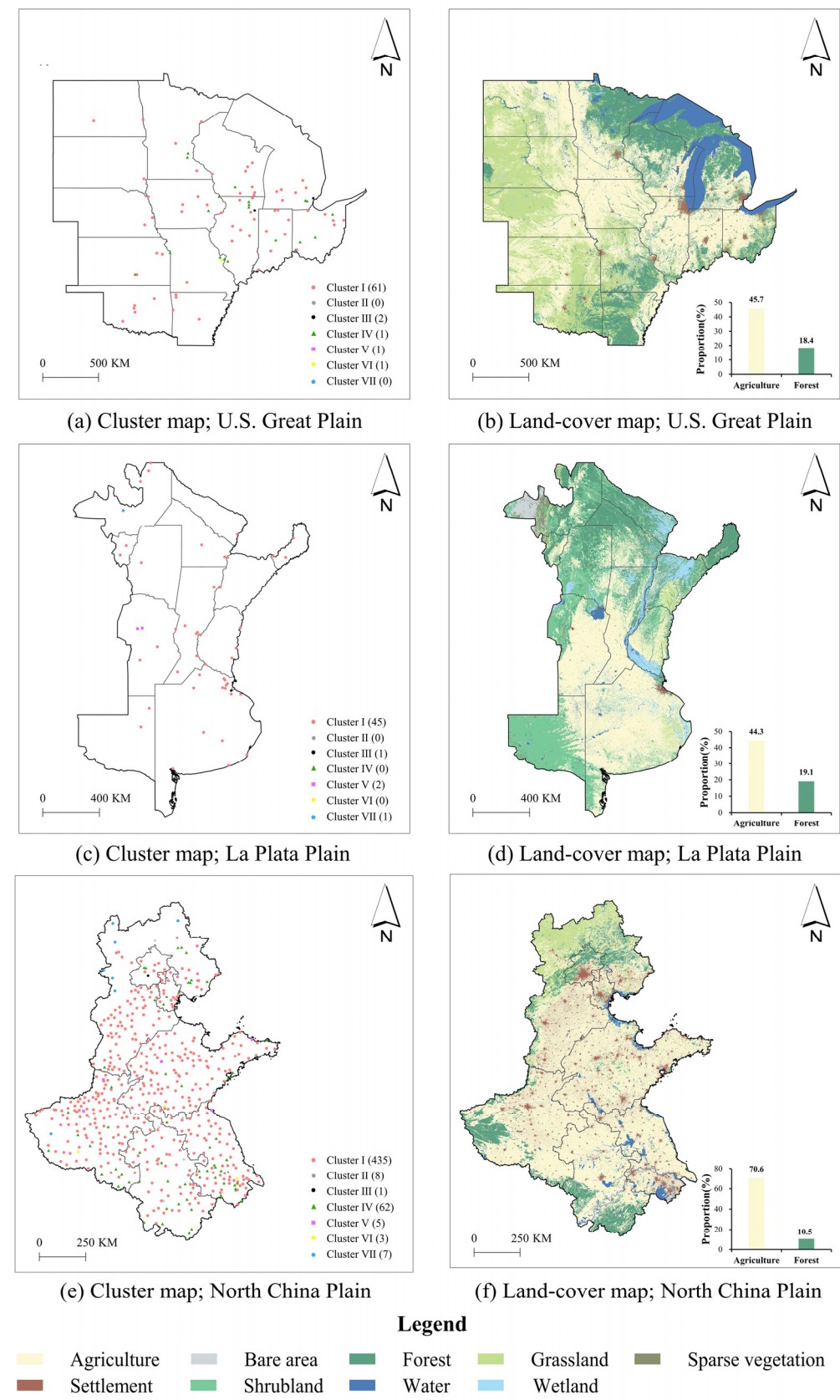

(a) Cluster map; U.S. Great Plain

(b) Land-cover map; U.S. Great Plain

(c) Cluster map; La Plata Plain

(d) Land-cover map; La Plata Plain

(e) Cluster map; North China Plain

(f) Land-cover map; North China Plain

**Legend**

| | | |
|---|---|---|
| Agriculture | Bare area | Forest | Grassland | Sparse vegetation |
| Settlement | Shrubland | Water | Wetland | |

## Appendix D

Comparing the spatial pattern of different clusters and land-cover maps, for geographical regions with a relatively large percentage of forest.

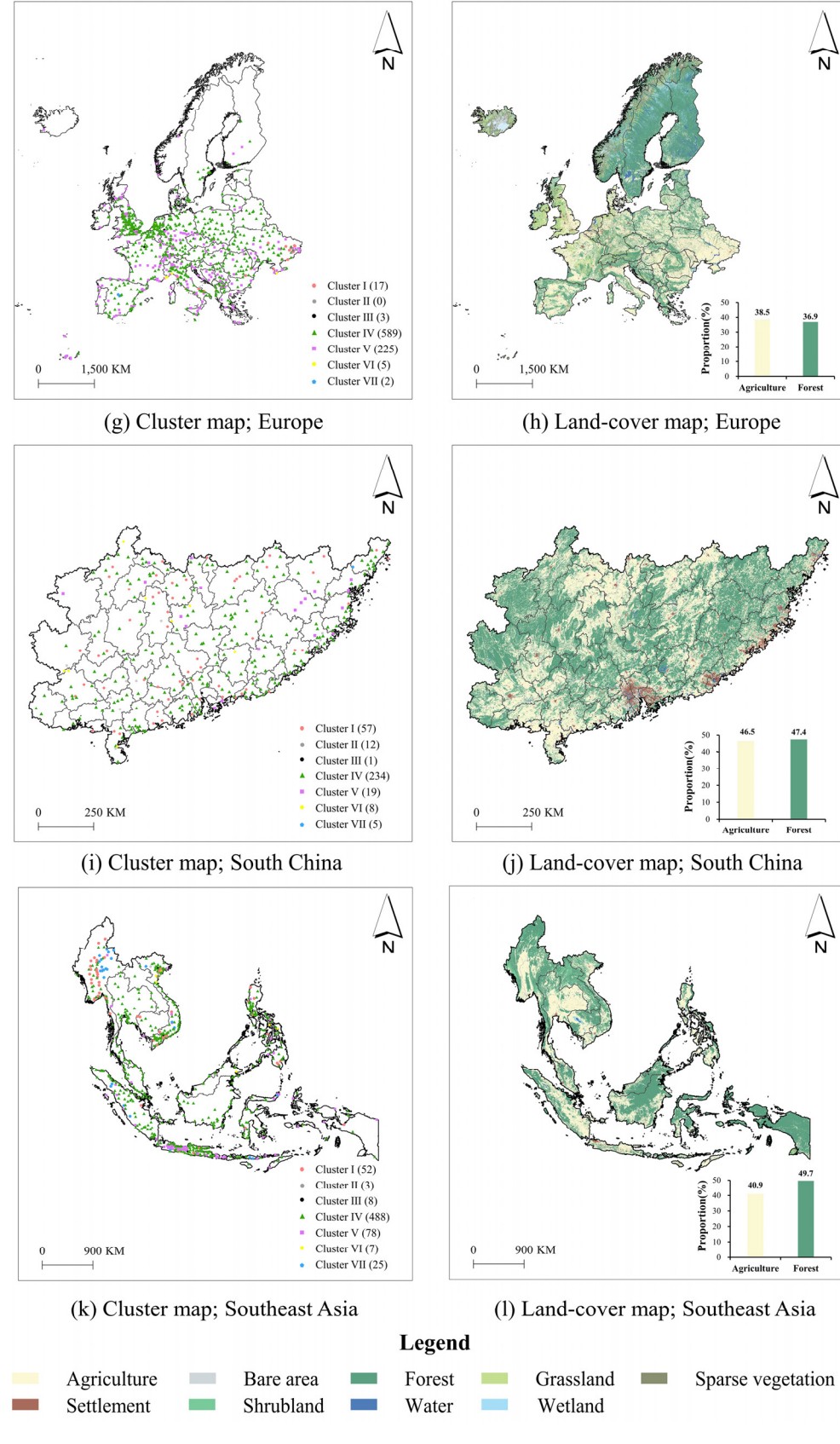

(g) Cluster map; Europe

(h) Land-cover map; Europe

(i) Cluster map; South China

(j) Land-cover map; South China

(k) Cluster map; Southeast Asia

(l) Land-cover map; Southeast Asia

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
