# Peer review of "Mapping Urban Forms Worldwide: An Analysis of 8910 Street Networks and 25 Indicators"

_ijgi, doi:10.3390/ijgi11070370_

Round 1

Reviewer 1 Report

See attached.

Author Response

Reviewer1

Reviewer1_ Comment1:

  1. The authors should specify why the six principal components are selected rather than 10 or 5.

Response to Reviewer1_ Comment1: Thanks for this comment!

The principle for selecting principle components are described in Section 3.2. That is,

Each principal component (PC) has a Eigenvalue to indicate the total amount of variance that can be explained by this PC. Commonly, the PCs whose Eigenvalues greater than one are kept, which means that these PCs can account for more variance than any original indicator did.

The corresponding references have also been added. That is,

Schwarz, N., (2010). Urban form revisited-Selecting indicators for characterising European cities. Landscape and Urban Planning, 96:29-47.

Sainani, K.L., (2014). Introduction to Principal Components Analysis. PM&R, 6(3):275-278.

Therefore, we selected six principle components because (see Section 4.2).

Six PCs have been found, and their Eigenvalues are all greater than one. These PCs can explain about 86% of the variance in the whole dataset.

Reviewer1_ Comment2:

  1. It is fine to define the name(s) of the categories of the PCs, but how could it be linked to literature on the factors influencing urban forms?

Response to Reviewer1_ Comment2: Thanks for this comment!

The six PCs were obtained through an analysis of the 25 urban form indicators published by Boeing (see Section 2.1). Thus the six PCs can still be highly correlated with one or several of the 25 urban form indicators (see Table 2). For instance, the PC1 has a strong positive correlation with several of these urban form indicators, e.g., area (0.899), built_up_area (0.881) and intersect_count (0.974). All these indicators are related to the size of a city and/or its street network. A high value of this PC indicates a relatively large size of a corresponding city. Thus this PC is denoted as 'Size' (this point has been added in Section 4.2).

Similarity, the names of the other PCs were also determined according to the strong correlation between each PC and the corresponding urban form indicators.

Reviewer1_ Comment3:

  1. Similarly, why do the authors choose to have seven clusters? There are many works on the classification of urban forms – the authors should have a more comprehensive literature review to help them develop the classification of the street patterns. Suggest reading, for instance, (https://doi.org/10.1098/rsif.2014.0924 and https://doi.org/10.1080/10630732.2021.2001713.)

Response to Reviewer1_ Comment3: Thanks for this comment!

First of all, the optimal number of clusters can be determined using the classical Elbow Method. This point has been added in the revised manuscript. That is (see Section 3.3),

Furthermore, the optimal number of clusters can be determined using the Elbow Method. The principle of this method is to calculate the Within-Cluster-Sum of Squared Errors (WCSS) for different numbers of clusters, and to identify the number of clusters that the WCSS dramatically decreases as the optimal one (Cui 2020).

Second, we have compared our clustering results with those in existing studies, and this point has been added in the discussion (please see Section 5.1). That is,

5.1. Comparing with existing studies

First of all, most of existing studies have used areal indicators (e.g., patches, blocks or cells) for the clustering of urban street networks [17,19,21]. However, in this study, we employed the indicators related to street nodes (e.g., intersect_count and prop_4way) and street segments (e.g., length_mean and circuity), which can be viewed as point and linear indicators, respectively. Thus our clustering results are different with those found in existing studies. More important, most of the existing studies have only focused on proposing indicators and approaches for the identification of urban form, but in our study, the relation between various street network forms and different geographical factors have also been investigated. Thus our study can be viewed as an extension and supplement of existing studies.

Furthermore, a large number (8910) of global street networks have been analyzed. They are much more than those (e.g., 100-231) in most of existing studies [17,19,21]. More important, with such a large sample size, it is possible to investigate the similarity and difference of street network form not only across different countries but also within a country. As an example, we found that the street networks in the United States may be characterized by an irregular pattern (e.g., Boston), but they were mostly recognized as a grid/regular pattern in previous studies [22]. Thus the clustering results also extend our understanding(s) of street network forms worldwide. Moreover, although several studies have also analyzed a large number of street networks [10,28-29], however, their clustering results are related to urban development [10] and urban sprawl [28-29]. In contrast, our clustering results are useful to understand the relation between street network forms and geographical factors on the global scale.

Third, the corresponding references have also been added. For instance (see Introduction),

  • Fang et al. proposed to use deep neural networks and planning guidance for the prediction of street configurations. (https://doi.org/10.1080/10630732.2021.2001713)
  • Louf and Barthelemy [19] used the area and shape of street blocks as indicators, and they divided 131 cities in the world into four clusters according to the size of area and the regularity of shape. (https://doi.org/10.1098/rsif.2014.0924)

Reviewer1_ Comment4:

  1. Moreover, the authors can highlight their original findings from this research. Are

there new categories / emerging urban forms compared with the existing studies?

Response to Reviewer1_ Comment4: Thanks for this comment! We have compared our clustering results with those in existing studies, and this point has been added in the discussion (please see Section 5.1). That is,

5.1. Comparing with existing studies

First of all, most of existing studies have used areal indicators (e.g., patches, blocks or cells) for the clustering of urban street networks [17,19,21]. However, in this study, we employed the indicators related to street nodes (e.g., intersect_count and prop_4way) and street segments (e.g., length_mean and circuity), which can be viewed as point and linear indicators, respectively. Thus our clustering results are different with those found in existing studies. More important, most of the existing studies have only focused on proposing indicators and approaches for the identification of urban form, but in our study, the relation between various street network forms and different geographical factors have also been investigated. Thus our study can be viewed as an extension and supplement of existing studies.

Furthermore, a large number (8910) of global street networks have been analyzed. They are much more than those (e.g., 100-231) in most of existing studies [17,19,21]. More important, with such a large sample size, it is possible to investigate the similarity and difference of street network form not only across different countries but also within a country. As an example, we found that the street networks in the United States may be characterized by an irregular pattern (e.g., Boston), but they were mostly recognized as a grid/regular pattern in previous studies [22]. Thus the clustering results also extend our understanding(s) of street network forms worldwide. Moreover, although several studies have also analyzed a large number of street networks [10,28-29], however, their clustering results are related to urban development [10] and urban sprawl [28-29]. In contrast, our clustering results are useful to understand the relation between street network forms and geographical factors on the global scale.

Reviewer1_ Comment5:

  1. The results will be more justifiable if the authors can find other statistics for validation. However, I also understand the difficulty to find a benchmark for those global studies. Perhaps the authors may think about whether some local validations can be implemented (e.g. in the next steps).

Response to Reviewer1_ Comment5: Thanks for this comment!

  • In terms of street network or urban form, we picked up 16 cities worldwide and visualized them for visual inspection (Please see Figure 3). These cities were chosen because they are not only located in some typical countries and regions mentioned in our study, but also close to corresponding cluster centroids.
  • In terms of the relationship between urban form and geographical landscape, First of all, an existing study has reported the quantitative relationship between urban form and geographical landscape factors (including terrain and land-cover). That is (see Introduction),

Zhou et al. [24] employed spatial autoregressive models to quantitatively understand the relation between street network form and multiple geographical factors. They found from an analysis of Chinese cities that not only the terrain but also various land-cover types (e.g., cultivated lands and forests) can be significantly correlated with street orientations.

Reference (it has also been added in the revised manuscript)

Zhou, Q., Lin, H., and Bao, J., (2021). Spatial autoregressive analysis of nationwide street network patterns with global open data. Environment and Planning B: Urban Analytics and City Science, 48(9): 2743-2760.

Second, our study is an extension of the existing study (Zhou et al. 2021). That is, we aim to investigate at a global scale that whether such urban forms can be related to geographical factors, which is achieved by involving global terrain and land-cover data into the analysis (this point has been highlighted in Introduction).

Reviewer 2 Report

The paper “Mapping urban forms worldwide: An analysis on 8910 street networks and 25 indicators” analyses the streets networks of 8910 cities in the world and define various clusters and possible links between the streets network and factors such as land cover and terrain elevation.

This work is well structured and the results are interesting even if, as declared by the author, there are some limitations in the used methodology that will be deepened in the future.

However, some aspects need to be modified before the paper can be published:

  • The text between lines 145 – 152 must be canceled because is repeated below.
  • Figure 2 a) is not clear, it could be modified or explained in the caption.
  • The statements in lines 369 – 374 and lines 379 - 392 should be justified by proper citations

Author Response

Reviewer2

Reviewer2_ Comment1:

1.The text between lines 145 – 152 must be canceled because is repeated below.

Response to Reviewer2_ Comment1: Thanks for this comment! The text between 145-152 has been removed from the revised manuscript.

Reviewer2_ Comment2:

  1. Figure 2 a) is not clear, it could be modified or explained in the caption.

Response to Reviewer2_ Comment2: Thanks for this comment! We have revised the caption (Figure 2a) into "hierarchical dendrogram".

Moreover, we have also explained what is "hierarchical dendrogram" in Section 3.3. That is,

The hierarchical clustering method is used for several reasons: First, this method can provide a hierarchical tree or dendrogram, which records the relationship of merges or splits during the clustering processing.

Reviewer2_ Comment3:

  1. The statements in lines 369 – 374 and lines 379 - 392 should be justified by proper citations

Response to Reviewer2_ Comment3: Thanks for this comment! These corresponding references have been added.

  1. David, G.S. and M. Tong, (2017). Transcending Type: Design for Urban Complexity. Urban Planning Forum, 2:50-60.
  2. Tong, M. , (2014) Urban Design and Theories in a Wider Scope. Urban Planning Forum, 1:50-60.
  3. McHarg, I. L. (1969). Design with nature. New York: American Museum of Natural History, pp. 7-17.
  4. Kostof, (1991). The City Shaped: Urban Patterns and Meanings through History, Bulfinch Press, Boston, MA.
  5. Sen, S. (2010). Between dominance, dependence, negotiation, and compromise: European architecture and urban planning practices in colonial India. Journal of Planning History, 9(4):2-3-231.

The statements have been justified by proper citations. That is,

  • “Revolution, cities could be interpreted as being made up of standardized components, which had changed the typology and terrain rooted in geography or culture[39]”
  • “Yet, cities are the product of historical processes, not only because of their physical nature, but also because of their cultural attributes[40].”
  • “In this sense, it is reasonable to follow the natural terrains while developing the city road networks [41]”
  • “In addition to the influence by physical geography, the irregularity can also be partly attributed to cultural elements[40,42].”
  • “This kind of idea was largely inherited by later developers and the idea of natural irregular pattern was also brought to India by early colonies from Europe[43].”

Reviewer 3 Report

This paper analyzes and clusters forms of road networks of 8910 cities across globe. The topic is interesting and the paper is easy to follow. I have the following comments:

  1. abstract: the results and implications from this paper have not been summarized properly. For example, “…. Six principal components have been found”: what components? These components should be important factors to show the urban form (I can see explanation from Section 4.2) but they were not delivered clearly in the abstract. Also, the implications and meanings of the results have not been elaborated.
  2. Table 1. Why the 25 indicators of road network are comprehensive? The authors need to provide more supporting evidence on the road network characteristics.
  3. Section 2.1. May need to provide a statistic about the dataset, e.g., the geographic distribution of the 8910 cities, e.g., Asian cities, American cities, or European cities.
  4. Figure 3. The sub-figure in the top should be numbered. Number of cities in each cluster needs to be presented.
  5. Figure 5 is confusing. The spatial correlation between road network clusters and distribution of “agriculture” and “forest” needs to be carefully examined instead of visual description. Moreover, evidences are needed to support the statement regarding the relationship between road network and the geographical characteristics.

Author Response

Reviewer3

Reviewer3_ Comment1:

  1. abstract: the results and implications from this paper have not been summarized properly. For example, “…. Six principal components have been found”: what components? These components should be important factors to show the urban form (I can see explanation from Section 4.2) but they were not delivered clearly in the abstract. Also, the implications and meanings of the results have not been elaborated.

Response to Reviewer3_ Comment1: Thanks for this comment!

  • First, the six principal components have been highlighted in the abstract, i.e., size, terrain-variation, regularity, long-street, circuity and altitude (see Abstract).
  • Second, the implications and meanings of the results have also been highlighted. That is (see Abstract),

Most of these clusters can be interpreted using terrain and land-cover data, which indicates that the urban forms of most cities across the globe are related to geographical factors. Thus the clustering results may be used not only to compare street networks and their urban forms at a global scale, but also to understand the formation and development of an urban street network.

Reviewer3_ Comment2:

  1. Table 1. Why the 25 indicators of road network are comprehensive? The authors need to provide more supporting evidence on the road network characteristics.

Response to Reviewer3_ Comment2: Thanks for this comment! We used the 25 indicators because the data were freely available. Specifically, we have highlighted this point in Section 2.1. That is,

The global urban street network data, produced by Boeing [12] and published in September 2020, are used for analysis. There are several advantages of using the data. First, the data include street networks of 8910 cities across the globe. Second, in the datasets, there are not only geometric data but also a dozens of form indicators for each street network. Third, the global urban street network data can be freely downloaded from the Harvard Dataverse.

There are other urban form indicators that may be considered. But, to the best of our knowledge, neither any data of such indicators, nor any open tool to calculate them are available. Therefore, these indicators were not considered in our study. Nevertheless, on the one hand, we compared the clustering results with several existing studies (see Section 5.1). On the other hand, we also highlighted in the discussion (see Section 5.3) that:

Besides, in additional to the 25 form indicators, it is also interesting to combine other indicators (e.g., fractal dimension, compact ratio and shape factor [20]) into analysis, in order to investigate the spatial pattern of global urban street networks in different perspective(s).

Reviewer3_ Comment3:

  1. Section 2.1. May need to provide a statistic about the dataset, e.g., the geographic distribution of the 8910 cities, e.g., Asian cities, American cities, or European cities.

 Response to Reviewer3_ Comment3: Thanks for this comment! A statistic table (see Table 1 in the revised manuscript) has been given out about the geographic distribution of the 8910 cities.

Table 1. A statistic of the 8910 cities across the globe

Region

Africa

Asia

Europe

North America

Oceania

South America

Number of cities

1504

4935

1051

372

41

1007

Proportion (%)

16.88

55.39

11.80

4.18

0.46

11.30

Reviewer3_ Comment4:

  1. Figure 3. The sub-figure in the top should be numbered. Number of cities in each cluster needs to be presented.

Response to Reviewer3_ Comment4: Thanks for this comment!

  • The sub-figure in the top has been numbered.
  • Besides, the number of cities in each cluster have also been presented.

Reviewer3_ Comment5:

  1. Figure 5 is confusing. The spatial correlation between road network clusters and distribution of “agriculture” and “forest” needs to be carefully examined instead of visual description. Moreover, evidences are needed to support the statement regarding the relationship between road network and the geographical characteristics.

Response to Reviewer3_ Comment5: Thanks for this comment!

  • First of all, we have added two figures (see Appendix C and D) to show the details of some typical regions in the world, and also quantitatively calculated the percentage of LC types (agriculture and forest) in these regions.
  • Second, an existing study has reported the quantitative relationship between urban form and geographical landscape factors (including terrain and land-cover). That is (see Introduction),

Zhou et al. [24] employed spatial autoregressive models to quantitatively understand the relation between street network form and multiple geographical factors. They found from an analysis of Chinese cities that not only the terrain but also various land-cover types (e.g., cultivated lands and forests) can be significantly correlated with street orientations.

Reference (it has also been added in the revised manuscript)

Zhou, Q., Lin, H., and Bao, J., (2021). Spatial autoregressive analysis of nationwide street network patterns with global open data. Environment and Planning B: Urban Analytics and City Science, 48(9): 2743-2760.

Round 2

Reviewer 3 Report

I do not have further comments.